# Investigation of Changes in Spermatozoon Characteristics, Chromatin Structure, and Antioxidant/Oxidant Parameters after Freeze-Thawing of Hesperidin (Vitamin P) Doses Added to Ram Semen

**DOI:** 10.3390/life12111780

**Published:** 2022-11-03

**Authors:** Deniz Yeni, Şükrü Güngör, Fatih Avdatek, Mehmet Fuat Gülhan, Kemal Tuna Olğaç, Muhammed Enes İnanç, Barış Denk, Umut Taşdemir

**Affiliations:** 1Department of Reproduction and Artificial Insemination, Faculty of Veterinary Medicine, Afyon Kocatepe University, Afyonkarahisar 03200, Turkey; 2Department of Reproduction and Artificial Insemination, Faculty of Veterinary Medicine, Mehmet Akif Ersoy University, Burdur 15030, Turkey; 3Department of Medicinal and Aromatic Plants, Vocational School of Technical Sciences, Aksaray University, Aksaray 68100, Turkey; 4Department of Reproduction and Artificial Insemination, Faculty of Veterinary Medicine, Ankara University, Ankara 06560, Turkey; 5Department of Biochemistry, Faculty of Veterinary Medicine, Afyon Kocatepe University, Afyonkarahisa 03200, Turkey; 6Department of Reproduction and Artificial Insemination, Faculty of Veterinary Medicine, Aksaray University, Aksaray 68100, Turkey

**Keywords:** semen, hesperidin, cryopreservation, oxidative stress, antioxidant, ram

## Abstract

We conducted this study to determine the potential cryopreservative effects of different hesperidin (vitamin P; H) doses on ram semen after freeze-thawing. Semen samples were obtained from Sönmez rams using an artificial vagina. The samples were divided into six groups: control, 10, 50, 100, 250, and 500 µg/mL H (C, H10, H50, H100, H250, and H500, respectively). At the end of the study, sperm motility and kinetic parameters, acrosome integrity (AI), mitochondrial membrane potential (MMP), viability, lipid peroxidation levels (LPL), chromatin damage, oxidant parameters, and antioxidant parameters were assayed. None of the doses of H added to the semen extender showed any enhancing effects on progressive motility compared to C (*p* > 0.05). In fact, H500 had negative effects (*p* < 0.05). Moreover, AI was the highest at the H10 dose, while LPL values were the lowest at the same dose (*p* < 0.05). The doses of H10 and H50 added to the Tris extender medium showed positive effects on sperm cell chromatin damage. Consequently, we can say that H doses used in this study are not effective on semen progressive motility, but the H10 dose is effective on AI and chromatin damage by reducing LPL.

## 1. Introduction

In Turkey, studies have been conducted on the crossbreeding of suitable genotypes for the breeding of domestic sheep. One of these domestic breeds is Sönmez. The Sönmez breed was obtained by crossing the Chios (25%) and Tahirova (75%) genotypes. To ensure the genetic continuity of these special breeds, the most practical technology for rapid genetic improvement at appropriate times is artificial insemination [1]. The success of this technique depends on the long-term protection of sperm cells from factors that affect their fertilization capacity negatively [2]. Cryopreservation has been developed with technological advancements and scientific studies and is still widely used today for the optimum preservation of the properties of cells and tissues. During the cryopreservation process, the freeze-thaw process may cause oxidative damage, decreased sperm cell motility, deterioration of sperm cell morphology, and consequently, a decrease in fertilization capacity [3]. Adding or removing substances with cryoprotectant properties to the semen may cause disruption in the intracellular osmotic balance and damage the genetic material of sperm cells [4]. Both seminal plasma and sperm cells are highly prone to lipid peroxidation (LPO) because they contain rich polyunsaturated fatty acids [5]. LPO triggers many chains of chemical reactions [6] and can damage important biological molecules such as proteins, DNA, and RNA [7]. Various enzymatic and non-enzymatic detoxification mechanisms attempt to eliminate the harmful effects of reactive oxygen species (ROS). These mechanisms are generally referred to as antioxidants [8].

Seminal plasma antioxidants play a crucial role in protecting spermatozoa against oxidative stress [9]. Oxidative stress in semen results from either the depletion of seminal antioxidants or excessive free radical formation by the sperm themselves. Therefore, increasing the antioxidant activity in the seminal plasma is extremely important in the cryopreservation of semen [10]. The application of antioxidants to semen diluents has received great interest in assisted reproductive technology applications. To protect sperm cells against various shocks during processing or storage, different diluents are designed, often on a testable basis. While natural additives added to diluents have been examined in the relevant literature, it was determined that there was no research on hesperidin (vitamin P; H) in the context of this subject. H (5,7,3′-trihydroxy-4′-methoxy flavanone), a flavonoid, is naturally found in citrus fruits such as lemons, oranges, and grapefruit [11]. Additionally, the protective effect of H on reproductive functions against toxicities caused by different chemicals has been reported by previous studies [12,13]. It was reported that H significantly reduces LPO in testicular tissue, has positive effects on sperm parameters and biochemical parameters, and improves epididymal functions [12]. Several mechanisms emerge to explain the biological effects of H on reproductive functions. The first is that it affects the secretion and activity of hormones. H was reported to have both estrogenic (at low concentrations) and anti-estrogenic (high concentrations) effects, depending on its concentration in the organism [14]. These hormones modulate the action mechanisms of hormones, such as the thyroid hormone [15]. Secondly, it was explained that H affects reproductive functions by inhibiting enzymes, such as aldose reductase, xanthine oxidase, phosphodiesterase, Ca^2+^-ATPase, lipoxygenase, and cyclooxygenase [16]. This study was designed to evaluate the protective effect of H on ram semen by reducing the damage that may occur after freeze-thawing and determine oxidative stress, chromatin damage, sperm motility, LPO levels, and certain antioxidant parameters.

## 2. Materials and Methods

### 2.1. Animals and General Experimental Procedure

In the study, semen samples were taken from six Sönmez breed rams (2–3 years old; Kapucuoğlu sheep farm, Afyonkarahisar, Turkey) using an artificial vagina. These procedures were carried out in the same breeding season and at different times as ten replications for each group. The ejaculates were used individually for each replication to eliminate the effects caused by different rams. A volumetric cup and a photometer were used to determine semen volume and concentrations. The mass activity (≥+++ 3 [scale 1–5]), sperm concentrations (≥0.8 × 10^9^/mL, volume ≥ 0.8 mL), and motility (80%) of the ejaculates were used to qualify the sources of semen. Tris (3.63 g Tris, 0.5 g fructose, 1.82 g citric acid, 100 mL double-distilled water, 7% glycerol (*v*/*v*) and 15% egg yolk) was used as the main diluent, and H was dissolved in dimethyl sulfoxide (DMSO; 1 g H, 1 mL DMSO). The extender osmolarity of the solutions was adjusted to 310 mOsm. After a preliminary study was done from the doses calculated based on the molecular weight of H, the treatment doses were decided. The semen samples were divided into six groups, and 0, 10, 50, 100, 250, and 500 µg/mL H were added separately to the groups (control (C), H10, H50, H100, H250, and H500, respectively). After these procedures, the samples that were injected (15 × 10^6^) into French straws were stored until they reached an equilibrium temperature (approximately 2 h at 4 °C). After providing the optimum cooling conditions, all samples were frozen at five programmed rates (0 °C/min from +4 °C to +4 °C 3 min, −3 °C/min from +4 °C to −7 °C 3.67 min, 0 °C/min from −7 °C to −7 °C 1 min, −37.67 °C/min from −7 °C to −120 °C 3 min, 0 °C/min from −120 °C to −120 °C 5 min). Afterwards, they were stored at −196 °C for 6 months. Ethics committee approval for the study was provided by Afyon Kocatepe University Faculty of Veterinary Medicine Animal Care Committee with the decision numbered 49533702/333.

### 2.2. Sperm Motility and Kinetic Parameters

The motility and kinetic parameters of ram semen were evaluated with a computer-assisted semen analyzer (CASA) in a phase contrast microscope using special software Sperm Class Analyzer (Microptics, Barcelona, Spain). Frozen semen was thawed at 37 °C for 30 s just before the evaluations. CASA was pre-adjusted for ram sperm analysis. A semen sample was extended 1:4 in the Lactated Ringer solution, and the extended semen samples were placed onto a pre-warmed 20-mm chamber slide (Leja 4, Leja Products BV, Nieuw-Vennep, The Netherlands). Characteristic analyses of motility were carried out with the aid of a green filtered negative phase contrast lens (100× magnification, at 37 °C). Curvilinear velocity variations for motile spermatozoa were categorized as fast (>75 µm/s), medium (45–75 µm/s), slow (10–45 µm/s), and static (<10 µm/s). Sperm with ≥75% flatness were considered progressive. In addition to these parameters, the values for kinetic calculations were quantified as follows: total motility (%), progressive motility (%), curvilinear velocity (VCL, µm/s), average path velocity (VAP, µm/s), straight line velocity (VSL, µm/s), amplitude lateral head displacement (ALH, µm/s), beat cross frequency (BCF, Hz), linearity (LIN %) [(VSL/VCL) × 100], yaw (WOB %) [(VAP/VCL) × 100], and straightness (STR %) [(VS/VAP) × 100]. In the evaluation of these parameters, a total of 200–400 sperm were recorded per sample in six microscopic zones.

### 2.3. Sperm Chromatin Damage

Chromatin damage in semen was analyzed using the single cell gel electrophoresis (COMET) assay method under intensively alkaline conditions. Semen was thawed at 37 °C in 30 s. The thawed semen was washed twice with PBS (Ca and Mg Free). Approximately 15 × 10^6^ (10 mL) washed sperm cells were mixed with low melting point agarose (LMA) at 37 °C. They were then spread on slides pre-coated with 1% normal soluble agarose (NMA). Lysis solution (Trevigen Inc. Cat. No. 4250-010-01) added to Triton X-100 (1%) in a vertical jar was immersed in the Triton X-100 and incubated at 4 °C for one h. Then, DL-Dithiothreitol (DTT) (4 mM) (Sigma Aldrich Chemical Co., Darmstadt, Germany) was added and incubated at 4 °C for another hour. Then, 60 µL of proteinase K (1 mg/mL) (Sigma Aldrich Chemical Co.) was added. The slides were then incubated at 37 °C overnight to dissolve the DNA. After being placed in an electrophoresis tank containing a previously prepared and cooled electrophoresis solution containing 300 mM NaOH and 1 mM Na-EDTA (pH 13), it was left to adapt to the solution for 15 min. An electric field (300 mA, 25 V) was applied for 20 min at 25 °C to pull the negatively charged DNA towards the anode. The slides were washed three times in neutralizing buffer (0.4 M Tris, pH 7.5) for 5 min at 25 °C and stained with 5 µg/mL Ethidium Bromide (Sigma-Aldrich Chemical Co.). They were then examined under a fluorescent microscope (Olympus CX31, Tokyo, Japan). Then, the Comet Score Freeware v1.5 software (The TriTek Co., Sumerduck, VA, USA) was used to evaluate the slides [17].

### 2.4. Flow Cytometric Analyses

The flow cytometric analyses were performed by using a CytoFLEX flow cytometer (Beckman Coulter, Brea, CA, USA) equipped with 525 ± 40 nm, 585 ± 42 nm, and 610 ± 20 nm emission filters and a 50-mW laser output (488 nm laser beam). All frozen semen was thawed at 37 °C 30 s just before the evolutions. In each analysis, ~10,000 events were examined. A side scatter area (SSC-A) versus forward scatter area (FSC-A) pseudo plot was used to exclude debris from the analysis, and duplicates were excluded by using forward scatter height (FSC-H) and forward scatter area (FSC-A) [18]. Working solutions of fluorescein isothiocyanate-conjugate peanut agglutinin (FITC-PNA) [100 µg/mL, Sigma, L7381], Sybr-14 and propidium iodide (PI) solution [1:10 Sybr-14, 2.99 mM PI, L7011, molecular probes, Invitrogen], 5,5′,6,6′-tetrachloro-1,1′3,3′-tetramethylbenzimidazolyl-carbocyanine iodide (JC-1) [0.153 mM T3198, molecular probes, Invitrogen], and BODIPY (5 µM, D38611, molecular probes, Invitrogen) were prepared with DMSO, divided into 50 µL portions, and stored at −20 °C until use.

Acrosome integrity (AI) values were determined using the FITC/PNA-PI staining kit according to the method described in a previous study [19]. The frozen-thawed sperm sample was diluted to 5 × 10^6^ sperm in 492 µL of PBS. Subsequently, 5 µL of FITC and 3 µL of PI were added, followed by incubation in a water bath at 37 °C for 15 min in a dark room. After incubation, the debris (non-sperm) was gated out, and sperm PMAI (FITC/PNA-PI-) analyses were performed using the CytExpert 2.3 software (Beckman Coulter, Brea, CA, USA)

Lipid peroxidation level (LPL) values were determined using BODIPY-SYBR staining according to the method reported in a previous study [20] with some modification. The frozen-thawed sperm sample was diluted to 5 × 10^6^ sperm in 492 µL of PBS. Subsequently, 5 µL of BODIPY and 3 µL of SYBR were added, followed by incubation in a water bath at 37 °C for 15 min in a dark room. After incubation, the debris (non-sperm) was gated out, and sperm cell LPL (BODIPY) analyses were performed using the CytExpert 2.3 software.

Sperm cell mitochondrial membrane potential (MMP) values were determined using 5,5′,6,6′ tetrachloro-1,1′3,3′-tetramethyl benzimidazolyl-carbocyanine iodide (JC-1). The frozen-thawed sperm sample was diluted to a concentration of 5 × 10^6^ sperm in 495 µL of PBS. Subsequently, 5 µL of JC-1 was added to the sample, followed by incubation in a water bath at 37 °C for 15 min in a dark room. After incubation, the debris (non-sperm) was gated out, and MMP analyses were performed using the CytExpert 2.3 software [21].

Sperm cell viability was determined using the SYBR and PI protocol reported in a previous study [19] with some modification. The frozen-thawed sperm sample was diluted to 5 × 10^6^ sperm in 492 µL of PBS. Subsequently, 5 µL of SYBR-14 and 3 µL of PI were added to the sperm sample, followed by incubation in a water bath at 37 °C for 15 min in a dark room. After incubation, the debris (non-sperm) was gated out, and sperm viability analyses were performed using the CytExpert 2.3 software.

### 2.5. Oxidant and Antioxidant Parameter Determination

For the determination of oxidative stress parameters after thawing (37 °C 30 s), the semen samples were washed three times with PBS by centrifugation at 800× *g* for 20 min with a refrigerated centrifuge to separate them from the diluent. The supernatant was then completed up to 0.5 mL with PBS. The specimens were taken to falcon tubes in ice for homogenization. The sonication treatment was repeated six times by keeping them in the ice for 30 s after a ten-second sonication process. The level of malondialdehyde (MDA), indicative of lipid peroxidation, was measured in accordance with a previously reported method [22]. In this method, lipid peroxides react with thiobarbituric acid and absorb at 532 nm. The amount of MDA was calculated in units of nmol/mL. According to Ellman’s method, Glutathione (GSH) was measured spectrophotometrically at 412 nm, and the amount was calculated in units of mg/dL [23]. We used a colorimetric test kit (Rel Assay Diagnostics, Gaziantep, Turkey) for the measurement of total antioxidant status (TAS; as mmol/L in 660 nm). In this method, the oxidized radical 2,20 -azino-bis-(3-ethylbenzothiazoline-6-sulfonic acid) (ABTS) in the kit is reduced by the antioxidant compounds in the samples examined, and color changes are observed [24]. A colorimetric test kit (Rel Assay Diagnostics, Gaziantep, TR) was used for the measurement of total oxidant status (TOS). The oxidation of the Fe^2+^ in the kit reduced to Fe3+ by oxidizing compounds was determined spectrophotometrically at 660 nm, and the results were calculated in units of μm/L. The oxidative stress index (OSI) was calculated according to the formula OSI = [(TOS)/(TAS × 100)]. TOS (as μmol/L in 660 nm).

### 2.6. Statistical Analysis

The number of repetitions in the study was ten. The homogeneity of variances was determined by using the obtained numeric data, the Shapiro-Wilk normality test, and Levene’s test. The results are expressed in tables as mean (X) ± standard deviation (SD). Spermatological parameters were modulated to the GLM procedure of SPSS 22.0 (SPSS Inc., Chicago, IL, USA). Post hoc testing was performed for identifying the sources of significant interactions between parameters. All statistical analysis results were interpreted with a minimum margin of error of 5% (*p* < 0.05), and this rate was considered statistically significant.

## 3. Results

### 3.1. Sperm Motility and Kinetic Parameters

Although the highest total motility results were obtained in H10 and H50, it was determined that the doses of H added to the sperm extender did not have a statistically significant preservation effect on progressive motility in comparison to the control group (Table 1; *p* > 0.05). Additionally, it should be noted that the dose of H500 had a negative effect on motility. Except for STR, spermatozoon kinetic parameters were found to be significantly different in all treatment groups (*p* < 0.05), but an advantageous result was not obtained compared to C (Table 1). In particular, VAP, VSL, and VCL values were found to be greater in the C group than those in the treatment groups. It was determined that the treatment did not have positive effects on kinetic parameters.

### 3.2. AI, MMP, Viability, and LPL Evaluation

Table 2 shows the results of the application of different H doses on AI, MMP, viability, and LPL after freeze-thawing in ram semen. These data indicated that AI and LPL were positively affected by the H10 dose (*p* < 0.05). No dose of H caused a positive change in MMP (*p* < 0.05). Additionally, the viability parameter indicated that the highest dose of H500 caused toxic effects, because it was above the threshold (Table 2).

### 3.3. Chromatin Damage Evaluation

According to the data, sperm tail length, tail DNA, and tail moment were preserved at the H10 and H50 doses compared to the control group. It was determined that chromatin damage increased at high doses of H (Table 3; *p* < 0.05).

### 3.4. Oxidant and Antioxidant Parameters

As seen in Table 4, there were no significant changes in TOS and OSI (*p* > 0.05). While it was expected that the high GSH activity at the H500 dose and the low MDA level at the H100 and H500 doses would positively affect the quality of semen in these groups, this expectation did not materialize.

## 4. Discussion

During the long- or short-term storage of semen, temperature, cooling rate, the chemical contents of diluents, the cryoprotectant ratio, ROS, and seminal plasma components are the main determinants affecting the viability of sperm cell [25]. In previous studies, it has been reported that sperm viability and motility, as well as the integrity of both the plasma membrane and acrosome, are adversely affected after thawing during the cryopreservation process [26]. The motility and kinetic parameters of the ram semen samples containing H doses added to Tris-egg yolk extender that were thawed are given in Table 1. In the present study, it was determined that H applied at low doses did not have a positive effect on progressive motility, while its application at high doses had a toxic effect (*p* > 0.05). In studies conducted with different species, it has been revealed that H improved some parameters, which contradicted our findings. Samie et al. [27] reported that H (50 mg/kg/day) administered to rats contributed to an increase in sperm counts and motility. In another study, 20 µM H treatment during the cryopreservation of human sperm significantly improved the motility ratios of sperm cells after cryopreservation [28]. Aksu et al. [29] also showed that the application of H (300 mg/kg/day) for 7 days against reproductive damage in male rats caused an increase in the percentage of sperm motility. In another study in contrast with our results, it was demonstrated that a 200 mg/kg/day dose of H had a protective effect on the reproductive system by reducing degenerative changes in sperm motility and density in male rats treated with methotrexate [30]. As a result of an investigation of the effects of H on the toxicity of an anticancer drug, cisplatin, in the reproductive system, it was reported that a dose of 7 mg/kg/day may be beneficial for reproductive functions against cisplatin-induced toxicity by causing an increase in sperm motility [12]. The common result of studies in which positive effects on motility have been determined is that the negative effects of various sources of toxicity on reproductive cells can be eliminated by dose-dependent supplements of H. In these studies, the main reason for the increase in sperm motility was thought to be decreased lipid peroxidation. Unlike the studies presented above, the reason why no positive improvement was observed especially in terms of progressive motility in our study was thought to be the effect of the cryoprotective substances already available in the diluent that was used.

The application of H10 showed a significant effect on AI. This finding was supported by the low LPL levels at the same dose, which showed that the mitochondrial structure was preserved at the H10 dose of semen post-thawing. LPL is an important parameter in determining mitochondrial oxidative damage. The experimental results showed that the H10 dose was effective in maintaining sperm viability by reducing mitochondrial LPL after thawing. This was probably because low doses (such as H10 and H50) reduced lipid peroxidation, and high doses added to the diluent failed to maintain the pH and osmotic pressure balance. Although the H10 dose showed protective effects against acrosome membrane defects, cell death could not be prevented at high doses (H250 and H500). The energy needed by sperm cells for their motility can be provided by high rates of glycolysis or oxidative phosphorylation [31]. H can be used as a good exogenous antioxidant agent for sperm cells because it appears to provide low AI and high MMP ratios for the protection of the plasma membrane and acrosome of sperm cells. It was reported that toxic agents, which have negative effects on sperm functions, disrupt mitochondrial function and cause the depolarization of the mitochondrial membrane [32]. This situation in the membrane potential causes the opening of the cytochrome gates escaping into the mitochondrial pores and cytoplasm, and thus, the initiation of apoptosis [33]. Studies have shown that the correlation between MMP and plasma membrane integrity is significant in the preservation of frozen human [34], bovine [35], and ram [21] semen. In this study, the numerical decrease in MMP in the frozen-thawed ram semen samples at doses other than H10 compared to group C showed parallelism with the results of other studies. It was stated that a decrease in temperature experienced during the freezing process of sperm causes some functional and structural damage in viability, reducing fertility in rams [36] and bulls [37]. Unlike our findings, İsmail et al. [38] reported that curcumin nanoparticles (100 μg) added to goat semen extender preserved sperm viability and plasma membrane integrity. In another study, Valipour et al. [28] found that 20 µM H treatment during the cryopreservation of human sperm significantly improved the viability of the sperm after thawing. It was stated that melatonin showed positive effects on viability by increasing oxidative phosphorylation and facilitating the transition to the mitochondrial permeability transition pore [39]. While curative results were obtained on AI and LPL at the H10 dose, the absence of a progressive effect on viability and PMP in all treatment groups should be noted. These results were interpreted as the insufficient reduction in oxidative stress in the environment.

Scientific data have shown that excessive ROS production leads to chromatin mutations and reduces ATP production, resulting in slowing sperm motility [40]. The preservation of the chromatin integrity of sperm cells is one of the success criteria in transferring genetic information to the next generations [41]. Oxidative degradation can cause chromatin base sequence degradation, fragmentation, and cross-link of proteins [42]. Structural damage to sperm cell chromatin adversely affects oocyte penetration and fertilization ability [43]. Sperm chromatin is very easily affected by external factors, so it must be packaged in a very dense state to protect DNA. In this process, oxidative stress caused by ROS production may prevent chromatin packaging and cause the peroxidation of the sperm plasma membrane [44]. Similar to our study, there are some studies reporting that chromatin damage is prevented by antioxidant agents in ram sperm to ensure cryopreservation after thawing [41,45]. In accordance with the conclusions reached here, Trivedi et al. [46] determined that H doses of 25, 50, and 100 mg/kg daily had a protective effect on tail length, tail moment, and tail DNA in rats in a testicular toxicity model. Vijaya Bharathi et al. [47] explained that glucosyl H, a modified form of H, had a protective effect against chromatin damage in the sperm cells of rats, in which reproductive toxicity was induced by increasing cellular antioxidant levels. In accordance with this study, 50 and 100 μg/mL thymoquinone added to Tris semen extender reduced chromatin damage in Sönmez rams in a previous study by our research group [48]. Additionally, the ability of H to stabilize the mitochondrial membrane was reported to protect sperm cells from apoptosis and increase mitochondrial functions [28]. There is some evidence that H has a protective effect against oxidative and nitrosative modifications in DNA induced by endogenous and exogenous ROS production [27,45,49]. Our results revealed important data on the effects of adding H during semen cryopreservation, such as the reduction or complete elimination of chromatin damage, depending on the doses of H.

Polyunsaturated fatty acids in the spermatozoon membrane are exposed to excessive ROS attacks during the freeze-thawing process, and thus, they are oxidized, which results in increased LPO [50]. Contrary to our findings, Valipour et al. [28] showed that the level of ROS formation increased during the freeze-thawing process, and 20 μM H treatment significantly reduced ROS levels. In another study, the authors observed that testicular damage caused by diabetes increased MDA levels, but with H (50 mg/kg/day) treatment, MDA decreased, GSH increased, and TAS increased simultaneously [27]. Trivedi et al. [44] stated that H (25, 50, and 100 mg/kg) treatment applied together with a chemotherapeutic agent, doxorubicin, significantly reduced MDA in rat sperm, increased GSH levels, and thus, significantly reduced oxidative stress. Unlike the findings obtained in this study, Helmy et al. [51] reported that H showed positive effects in regulating the antioxidant capacity of testicular tissue and reducing cell death, testicular histology, and oxidative damage indicators. In another study, it was shown that H regulated testicular insults and had a protective effect against oxidative stress in an experimental varicocele model [52]. However, our findings revealed that the GSH levels of the semen samples after thawing did not show a significantly effective activity at any dose of H. Additionally, we cannot say that the H doses that were used in our study had a significant effect on TAS levels compared to the control group. The reason for this may be that these selected doses were not sufficient for an endogenous enzyme activity or in the context of the semen samples of the selected animal species. In line with our results, Kaneko et al. [53] reported that low GSH levels may reduce spermatogenesis and TAS levels. Similarly, a study on human semen reported no correlation between seminal parameters and ROS [54]. There was a negative relationship between oxidative stress and sperm cell quality [55]. It was observed that antioxidant molecules, especially low-weight molecules in the seminal plasma, can freeze and preserve semen better [56]. Kovalski et al. [57] emphasized the importance of select substances with low molecular weight in non-enzymatic exogenous antioxidant applications. It was shown that H, the exogenous antioxidant agent chosen for this study, plays a relatively minor role in increasing the antioxidant levels of semen in cryopreservation, depending on its molecular weight.

## 5. Conclusions

The positive and negative effects of many endogenous and exogenous antioxidant substances on cryopreservation have been documented by scientists in many regions of the world. However, some plant extracts used in studies may have harmful effects on reproductive functions, even if they have antioxidant activities. Therefore, it is important to elucidate the bioactive components, detailed chemical structures, animal species differences, and effective doses of such agents. In this study, H that was added to ram semen was directly supplied in pure form to determine its cryopreservative effects. For this reason, the results of our study on the cryopreservative effects of H at the determined doses revealed clearer information. Our results suggested that H doses are not very effective on semen progressive motility, but the H10 dose is effective in protecting AI and preventing chromatin damage by reducing lipid peroxidation levels. It was considered that it shows this effect by eliminating oxidative stress products in the environment in a non-enzymatic manner.

## Figures and Tables

**Table 1 life-12-01780-t001:** Sperm motility and kinetic parameters.

Parameters	C	H10	H50	H100	H250	H500	*p*
Prog. mot. (%)	13.73 ± 1.80 ^a^	16.99 ± 3.87 ^a^	16.08 ± 0.96 ^a^	13.13 ± 1.25 ^a^	11.35 ± 1.85 ^ab^	7.34 ± 0.83 ^b^	*
Total mot. (%)	51.92 ± 3.68 ^c^	67.70 ± 1.32 ^a^	63.59 ± 3.23 ^ab^	55.85 ± 3.72 ^bc^	53.85 ± 4.86 ^bc^	38.14 ± 3.80 ^d^	*
VAP (µm/s)	44.31 ± 4.89 ^a^	36.52 ± 0.99 ^b^	34.85 ± 2.07 ^b^	33.94 ± 1.63 ^b^	38.51 ± 1.67 ^ab^	31.96 ± 1.44 ^b^	*
VSL (µm/s)	26.85 ± 3.16 ^a^	22.24 ± 0.85 ^b^	20.39 ± 1.32 ^b^	20.87 ± 1.31 ^b^	23.94 ± 0.82 ^ab^	19.99 ± 1.54 ^b^	*
VCL (µm/s)	64.45 ± 4.42 ^a^	58.21 ± 0.98 ^b^	56.97 ± 2.12 ^bc^	53.17 ± 1.63 ^bc^	57.59 ± 1.89 ^bc^	51.47 ± 1.21 ^c^	*
ALH (µm/s)	2.37 ± 0.11^ab^	2.43 ± 0.07 ^a^	2.43± 0.08 ^a^	2.15 ± 0.03 ^b^	2.19 ± 0.06 ^ab^	2.18 ± 0.10 ^ab^	*
BCF (Hz)	8.76 ± 0.99 ^ab^	8.19 ± 0.26 ^ab^	8.04 ± 0.22 ^ab^	8.51 ± 0.29 ^ab^	9.14 ± 0.32 ^a^	7.73 ± 0.40 ^b^	*
LIN (%)	41.71 ± 2.63 ^a^	37.26 ± 1.17 ^ab^	35.57 ± 1.37 ^b^	37.81 ± 1.48 ^ab^	40.58 ± 0.83 ^ab^	38.41 ± 2.79 ^ab^	*
STR (%)	59.20 ± 1.37	57.71 ± 0.73	56.84 ± 0.54	58.20 ± 0.93	59.38 ± 0.83	59.15 ± 2.19	-
pOB (µm s^−1^)	66.71 ± 2.99 ^a^	61.67 ± 1.23 ^ab^	59.97 ± 1.86 ^b^	61.85 ± 1.54 ^ab^	65.11 ± 0.83 ^ab^	60.86 ± 2.28 ^b^	*

^a, b, c, d^ Different superscripts within the same row demonstrate significant differences (* *p* < 0.05). - No significant difference (*p* > 0.05), mean (±SE).

**Table 2 life-12-01780-t002:** AI, MMP, viability, and LPL values.

Parameters	C	H10	H50	H100	H250	H500	*p*
AI (%)	26.20 ± 2.36 ^b^	32.78 ± 1.82 ^c^	23.56 ± 1.43 ^ab^	25.94 ± 1.10 ^b^	20.67 ± 0.43 ^b^	13.66 ± 0.99 ^a^	*
MMP (%)	30.11 ± 2.88 ^c^	34.45 ± 1.36 ^c^	20.30 ± 2.16 ^ab^	15.17 ± 3.70 ^ab^	21.64 ± 2.60 ^b^	13.27 ± 0.21 ^a^	*
Viability (%)	62.50 ± 3.07 ^b^	66.39 ± 0.97 ^b^	61.52 ± 1.46 ^b^	60.74 ± 1.03 ^b^	57.12 ± 2.70 ^b^	43.92 ± 8.72 ^a^	*
LPL (%)	77.97 ± 7.91 ^bc^	64.95 ± 2.78 ^a^	71.21 ± 3.23 ^b^	73.64 ± 6.21 ^b^	85.09 ± 9.85 ^bc^	94.29 ± 1.58 ^c^	*

^a, b, c^ Different superscripts within the same row demonstrate significant differences (* *p* < 0.05), mean (±SE).

**Table 3 life-12-01780-t003:** Chromatin damage values.

Parameters	C	H10	H50	H100	H250	H500	*p*
Tail length (µm/s)	25.23 ± 0.92 ^b^	14.00 ± 0.99 ^d^	14.07 ± 0.83 ^d^	18.46 ± 1.29 ^c^	25.12 ± 1.12 ^b^	35.03 ± 1.16 ^a^	*
Tail DNA (%)	36.67 ± 3.13 ^b^	23.75 ± 2.17 ^a^	22.72 ± 2.40 ^a^	34.70 ± 0.96 ^b^	39.00 ± 2.01 ^b^	58.42 ± 3.57 ^a^	*
Tail moment (µm/s)	20.66 ± 1.86 ^b^	15.51 ± 1.78 ^cd^	12.36 ± 1.05 ^d^	18.91 ± 0.75 ^bc^	23.22 ± 1.18 ^b^	35.67 ± 2.06 ^a^	*

^a, b, c, d^ Different superscripts within the same row demonstrate significant differences (* *p* < 0.05), mean (±SE).

**Table 4 life-12-01780-t004:** GSH, MDA, TAS, TOS, and OSI values.

Parameters	C	H10	H50	H100	H250	H500	*p*
GSH (mg/dL)	37.30 ± 0.83 ^ab^	35.47 ± 0.45 ^a^	37.46 ± 2.11 ^ab^	36.26 ± 0.45 ^ab^	37.30 ± 0.48 ^ab^	39.36 ± 1.17 ^b^	*
MDA (nmol/mL)	4.63 ± 0.08 ^a^	4.42 ± 0.12 ^ab^	4.45 ± 0.11 ^ab^	4.15 ± 0.10 ^b^	4.39 ± 0.10 ^ab^	4.25 ± 0.06 ^b^	*
TAS (mmol/L)	17.02 ± 0.54 ^ab^	17.20 ± 2.19 ^ab^	17.24 ± 1.35 ^ab^	16.22 ± 0.51 ^a^	19.61 ± 1.28 ^b^	17.09 ± 1.10 ^ab^	*
TOS (μmol/L)	8.66 ± 0.31	9.32 ± 0.31	9.07 ± 0.22	9.40 ± 0.24	9.17 ± 0.13	9.35 ± 0.18	-
OSI	5.12 ± 0.29	5.49 ± 0.35	5.52 ± 0.66	5.81 ± 0.17	4.74 ± 0.21	5.62 ± 0.48	-

^a, b^ Different superscripts within the same row demonstrate significant differences (* *p* < 0.05). - No significant difference (*p* > 0.05), mean (±SE).

## Data Availability

The authors agree to share the collected data and statistical code upon reasonable request.

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
