# Peer review of "Investigation of Changes in Spermatozoon Characteristics, Chromatin Structure, and Antioxidant/Oxidant Parameters after Freeze-Thawing of Hesperidin (Vitamin P) Doses Added to Ram Semen"

_life, 2022, doi:10.3390/life12111780_

Round 1
Reviewer 1 Report
This is a quite interesting study of the effect of HSP addition ram sperm cryopreservation. Despite several sperm parameters being analysed the experimental design is not clear, since authors do not clearly indicate when HSP was added to the sperm samples. A detailed protocol of the procedure followed to freeze and thaw sperm samples has not been provided. Moreover, it is not clear when sperm samples were analysed after thawing.
Other specific comments that authors must address are detailed below:
Lines 31-32. Please, refer to “sperm chromatin damage” instead of“spermatozoon chromatin damage”.
Subsection 2.1. Animals and general experimental procedure. Please, provide a detailed description of the freeze-thawing procedure followed. In this section authors must clearly indicate in which step of the protocol HSP was added.
Subsection 2.4. Flow cytometric evaluation. In this subsection authors must indicate for each cytometric parameter which cell populations were identified in the cytometric plots and which sperm population was used to express the results.
The Results section must be improved by highlighting the most relevant results obtained from the analysis of each sperm parameter.
Table 1, Table 2, Table 3, and Table 4. Please provide an appropriate title for all these tables. Please note that “Mean (±SE)” must be placed in the table food and so all the abbreviations.
Table 4 shows the results obtained form Comet assay. According to the information given in this Table this assay provides three parameters: Tail length (mm/s), Tail DNA (%) and Tail moment (mm/s). In order to improve the comprehension of this table authors must clearly indicate in the Material and Methods section (subsection 2.3. Chromatin damage) how these parameters were obtained.
The Discussion section needs deep revision in to improve its quality. Some paragraphs are a mere repetition of the results obtained. Authors do not compare their results with those obtained in similar studies, but they just describe the results obtained by other researchers without highlighting the biological significance of similarities and of differences between studies. Authors must highlight the implications in the improvement of freeze-thawing protocols.
Similarly, the Conclusions sections needs deep revision in order to highlight the biological significance of the results obtained.
The Reference list is quite dated; authors must include recent references.
Author Response
Dear Reviewer;
The suggestions of you were answered one by one, and the added and revised sections are shown on the manuscript in red clour. Sections deleted from the manuscript are shown in blue and crossed out. After the revision, the manuscript was made to proofreading again. The proofreading statement can be provided upon request.
Response to Reviewer 1 Comments
Point 1: This is a quite interesting study of the effect of HSP addition ram sperm cryopreservation. Despite several sperm parameters being analysed the experimental design is not clear, since authors do not clearly indicate when HSP was added to the sperm samples. A detailed protocol of the procedure followed to freeze and thaw sperm samples has not been provided. Moreover, it is not clear when sperm samples were analysed after thawing.
Response 1: After a preliminary study was done from the doses calculated based on the molecular weight of H, the application doses were decided. The extenders were prepared separately at the specified doses for all treatment and control groups. Detailed protocols of sperm freezing and evaluation were written in the related sections. Frozen semen was thawed at 37°C 30 sec. just before the analysis and evaluation.
Point 2: Lines 31-32. Please, refer to “sperm chromatin damage” instead of“spermatozoon chromatin damage”.
Response 2: In line with the suggestion and necessary correction was done.
Point 3: Subsection 2.1. Animals and general experimental procedure. Please, provide a detailed description of the freeze-thawing procedure followed. In this section authors must clearly indicate in which step of the protocol HSP was added.
Response 3: In line with the suggestion, animals and general experimental procedure are presented in detail in the material method section. The extender was prepared separately for each treatment dose of hesperidin. It was extended after semen collection and pre-assessment.
Point 4: Subsection 2.4. Flow cytometric evaluation. In this subsection authors must indicate for each cytometric parameter which cell populations were identified in the cytometric plots and which sperm population was used to express the results.
Response 4: In line with the suggestion, flow cytometric evaluation are presented in detail in the material method section.
Point 5: The Results section must be improved by highlighting the most relevant results obtained from the analysis of each sperm parameter.
Response 5: In line with the suggestion, result section was improved.
Point 6: Table 1, Table 2, Table 3, and Table 4. Please provide an appropriate title for all these tables. Please note that “Mean (±SE)” must be placed in the table food and so all the abbreviations.
Response 6 : Table titles were changed in line with the suggestion and necessary corrections were done.
Point 7: Table 4 shows the results obtained form Comet assay. According to the information given in this Table this assay provides three parameters: Tail length (mm/s), Tail DNA (%) and Tail moment (mm/s). In order to improve the comprehension of this table authors must clearly indicate in the Material and Methods section (subsection 2.3. Chromatin damage) how these parameters were obtained.
Response 7: In line with the suggestion, COMET asssay is presented in detail in the material method section.
Point 8: The Discussion section needs deep revision in to improve its quality. Some paragraphs are a mere repetition of the results obtained. Authors do not compare their results with those obtained in similar studies, but they just describe the results obtained by other researchers without highlighting the biological significance of similarities and of differences between studies. Authors must highlight the implications in the improvement of freeze-thawing protocols.
Response 8: In line with the suggestion, discussion section was revised, repetition informations are deleted and shown in blue colour.
Point 9: Similarly, the Conclusions sections needs deep revision in order to highlight the biological significance of the results obtained.
Response 9: In line with the suggestion, conclusion section was revised
Point 10: The Reference list is quite dated; authors must include recent references.
Response 10 : In line with the suggestion, since there is no study on hesperidin semen freezing in recent studies. However, current references on the subject have been added.

Reviewer 2 Report
The article by Yeni and collaborators aimed to evaluate the protective action of hesperidin in the cryoprotectant of ram semen. The study idea is good, even with published articles on the same subject. I have some specific comments regarding the article title and the methods.
The title is a little confusing. It seems that only the hesperidin was frozen and thawed. Maybe the authors can improve to demonstrate the real objective of the article.
Please change the acronym HSP for hesperidin. This could be very confusing for the reader, once this is the name of the heat shock proteins.
Where do the authors were based to create the hesperidin doses? This should be reported in the method section.
The analyses that were performed in the flow cytometry should be subitems on topic 2.4. Otherwise, it can confuse the reader.
Please add detailed information about the comet assay.
Why do the authors create a new name for sperm acrosome integrity? Spermatozoon plasma membrane acrosome integrity does not make any sense. The same happened to mitochondrial membrane potential. The JC1 evaluates both high and low potential.
Please add detailed information of MDA, GSH, TAS ans TOS.
Author Response
Dear Reviewer;
The suggestions of you were answered one by one, and the added and revised sections are shown on the manuscript in red clour. Sections deleted from the manuscript are shown in blue and crossed out. After the revision, the manuscript was made to proofreading again. The proofreading statement can be provided upon request.
Response to Reviewer 2 Comments
Point 1: The article by Yeni and collaborators aimed to evaluate the protective action of hesperidin in the cryoprotectant of ram semen. The study idea is good, even with published articles on the same subject. I have some specific comments regarding the article title and the methods.
The title is a little confusing. It seems that only the hesperidin was frozen and thawed. Maybe the authors can improve to demonstrate the real objective of the article.
Response 1: We generally do oxidative studies. The titles of our previous publications were written on the effects of the antioxidant substance we used. With the idea of ​​giving a different title in this publication, we found the title presented appropriate for our study. We want the title to stay the same if it's okay with you.
Point 2: Please change the acronym HSP for hesperidin. This could be very confusing for the reader, once this is the name of the heat shock proteins.
Response 2: In line with the suggestion, the abbreviation HSP for hesperidin was changed to H.
Point 3: The analyses that were performed in the flow cytometry should be subitems on topic 2.4. Otherwise, it can confuse the reader.
Response 3: Flow cytometry analyzes are presented in a clear way under a single heading in line with your suggestion.
Point 4: Please add detailed information about the comet assay.
Response 4: In line with the suggestion, COMET asssay is presented in detail in the material method section.
Point 5: Why do the authors create a new name for sperm acrosome integrity? Spermatozoon plasma membrane acrosome integrity does not make any sense. The same happened to mitochondrial membrane potential. The JC1 evaluates both high and low potential.
Response 5: In line with the suggestion, The necessary corrections are made and shown on the manuscript.
Point 6: Please add detailed information of MDA, GSH, TAS ans TOS.
Response 6 : In line with the suggestion, oxidative stress analyzes are presented in detail.

Reviewer 3 Report
The MS “Investigation of changes in spermatozoa characteristics, chromatin structure, and antioxidant/oxidant parameters after freeze-thawing of hesperidin (vitamin P) doses added to ram semen” is not falls well within the scope of Special Issue. Therefore, in my opinion the MS is not suitable for publication.
General Comments
Abstract:
The abstract should add a sentence about the main functions of hesperidin.
L29 what is PMI?
Discussion
Please explain the difference of spermatozoon, spermatozoa and sperm; and consider whether to write uniformly in entire MS.
Author Response
Dear Reviewer;
The suggestions of you were answered one by one, and the added and revised sections are shown on the manuscript in red clour. Sections deleted from the manuscript are shown in blue and crossed out. After the revision, the manuscript was made to proofreading again. The proofreading statement can be provided upon request.
Response to Reviewer 3 Comments
Point 1: The MS “Investigation of changes in spermatozoa characteristics, chromatin structure, and antioxidant/oxidant parameters after freeze-thawing of hesperidin (vitamin P) doses added to ram semen” is not falls well within the scope of Special Issue. Therefore, in my opinion the MS is not suitable for publication.
Response 1: In line with suugestion, it has been accepted to send publications to the regular issue instead of the special issue related to this subject.
Point 2: The abstract should add a sentence about the main functions of hesperidin.
Response 2: Since the referees do not want to have literature information in the abstract section in general, the function of the hesperidin we mentioned in the introduction section is not specified separately in the abstract section.
Point 3: L29 what is PMI?
Response 3: The acronym PMI has been corrected to acrosome integrity (AI).
Point 4: Please explain the difference of spermatozoon, spermatozoa and sperm; and consider whether to write uniformly in entire MS.
Response 4: In line with the suggestion, spermatozoa/spermatozoon/sperm were changed to a sperm cells to provide a uniformity in the manuscript.

Round 2
Reviewer 1 Report
The authors have introduced all the modifications requested by this reviewer, and so the overall quality of the manuscript has been improved.
Reviewer 3 Report
The manuscript has met the requirements for publication.